# Therapeutic Drug Monitoring of Carbamazepine: A 20-Year Observational Study

**DOI:** 10.3390/jcm10225396

**Published:** 2021-11-19

**Authors:** Grzegorz Grześk, Wioleta Stolarek, Michał Kasprzak, Elżbieta Grześk, Daniel Rogowicz, Michał Wiciński, Marek Krzyżanowski

**Affiliations:** 1Department of Cardiology and Clinical Pharmacology, Faculty of Health Sciences, Collegium Medicum, Nicolaus Copernicus University, Skłodowskiej-Curie 9, 85-094 Bydgoszcz, Poland; ggrzesk@cm.umk.pl (G.G.); rogowicz.d@gmail.com (D.R.); 2Department of Pharmacology and Therapeutics, Faculty of Medicine, Collegium Medicum, Nicolaus Copernicus University, Skłodowskiej-Curie 9, 85-094 Bydgoszcz, Poland; wicinski4@wp.pl (M.W.); marekwjk@cm.umk.pl (M.K.); 3Department of Cardiology and Internal Diseases, Faculty of Medicine, Collegium Medicum, Nicolaus Copernicus University, Skłodowskiej-Curie 9, 85-094 Bydgoszcz, Poland; medkas@o2.pl; 4Department of Pediatrics, Hematology and Oncology, Faculty of Medicine, Collegium Medicum, Nicolaus Copernicus University, Skłodowskiej-Curie 9, 85-094 Bydgoszcz, Poland; ellag@cm.umk.pl

**Keywords:** carbamazepine, antiepileptic drugs, carbamazepine plasma levels, epilepsy, therapeutic drug monitoring

## Abstract

Background: Carbamazepine (CBZ) is a first-generation anticonvulsant drug. Hence, in certain cases, therapeutic drug monitoring (TDM) supports pharmacotherapy. Methods: The presented research was based on a retrospective analysis including 710 ambulatory and hospitalized patients treated with CBZ between the years 1991 and 2011. The method used for the determination of the CBZ concentration was fluorescence polarization immunoassay (FPIA) performed using an Abbott GmbH TDx automatic analyzer, with the therapeutic range for carbamazepine being 4–12 µg/mL. Results: The therapeutic range was observed more often in patients between 3 and 17 years of age compared with the population ≥18 years of age (73.5% vs. 68.8%). The therapeutic level was exceeded less frequently in the population between 3 and 17 years of age despite them being given a significantly higher dose per kilogram of body weight than in the population ≥18 years of age (13.64 mg/kg vs. 10.43 mg/kg, *p* < 0.0001). Patients ≥18 years of age were statistically significantly more likely to be in the group with a suspected drug overdose (73.9% vs. 26.1%), and suicide attempts only occurred in elderly patients (100.0% vs. 0.0%, *p* = 0.003). Conclusion: The results of the TDM of CBZ showed that only 71% of all samples were at the therapeutic level. To ensure the maximum efficacy and safety of the therapy, it is necessary to monitor the concentration of CBZ regardless of sex and age.

## 1. Introduction

Carbamazepine (CBZ), one of the classic antiepileptic drugs (AEDs), was introduced in the 1960s and 1970s. It still plays a dominant role in epilepsy treatment as the first-line drug of choice (DOC), especially for different focal epileptic syndromes [1,2,3]. In addition, this drug is used, among other things, to treat idiopathic neuralgias of the trigeminal and glossopharyngeal nerves and pain caused by diabetic neuropathy. Carbamazepine blocks voltage-gated sodium channels, thus stabilizing the cell membranes of nerve fibers, inhibiting neuronal discharge, and reducing excitatory synaptic transmission [4]. Pharmacokinetics in patients above 3 years of age are comparable to those in adults [5]. The dosage of CBZ for epilepsy in adults is 0.8–1.2 g/day 2–3 times a day. However, in children from 1 to 15 years of age, it amounts to 0.1–1 g/day (10–25 mg/kg/day) [6]. CBZ is a drug with a strong effect on metabolic processes that may cause clinically significant drug interactions.CBZ induces CYP3A4 and CYP1A2 isoenzymes, thus reducing the concentration of other drugs (e.g., DSA). CBZ is metabolized by CYP1A2, CYP3A4, CYP2C8, and CYP2C9 isoenzymes, and itself is also subject to interactions changing its serum concentration. The increase in CBZ concentration is caused by, inter alia, clarithromycin and erythromycin, while the reduction in CBZ concentration is caused by, among other things, phenobarbital and rifampicin [4]. Therapeutic drug monitoring (TDM) is recommended for the rapiesusing antiepileptic drugs, including CBZ. The posology can be optimized by TDM due to the good correlation between the plasmatic concentration and the therapeutic effect [6]. The test should be ordered individually in particular situations (e.g., in the case of drug interactions, for suspected drug poisoning, or for monitoring compliance with medical recommendations) [1,7]. It has been shown that the use of TDM in epilepsy treatment can improve seizure control and minimize the risk of side effects as well as limiting polytherapy, thus reducing treatment costs [8]. It should be remembered that drug side effects do not have to be associated with a high drug concentration in the plasma. On the other hand, the symptoms of poisoning can be misinterpreted as the symptoms of a primary disease of the nervous system.

Epilepsy is the most common neurological disease, with a prevalence ranging from 0.5% to 1% in developed countries [9]. Convulsions arise as a result of an imbalance between the excitatory and inhibitory systems in the central nervous system. The etiology of epilepsy is not fully understood (it may be the result of, among other things, perinatal trauma, infections of the central nervous system, head injury, and brain tumors) [10]. In order to effectively treat epilepsy, it is necessary to correctly diagnose the disease and introduce appropriate treatment. In approximately 70% of patients, monotherapy is the primary treatment method and allows full seizure control. The choice of an antiepileptic drug depends on the individual patient’s characteristics. The facts that should be taken into consideration here are the type of epilepsy, the patient’s age, the existence of comorbidities, and the possible drug interactions [11].

## 2. Materials and Methods

### 2.1. Patients

A CBZ level assessment was performed in the Department of Pharmacology and Therapy in patients from the Kuyavian-Pomeranian Voivodeship between the years 1991 and 2011. The trial was designed as a retrospective analysis and included 710 ambulatory and hospitalized patients. The determination of the CBZ level was most often combined with consultation with a clinical pharmacologist. At that time, it was the only such center in the region that measured the concentration of both antiepileptic (CBZ, valproic acid, phenytoin) and other drugs, e.g., digoxin [12]. Moreover, it was also the only center in the voivodeship that measured the concentration of novel oral anticoagulants (NOACs) in the following years [13]. The study population took carbamazepine in the form of oral tablets.

### 2.2. Procedures

For all patients, the carbamazepine concentration in the blood serum (without anticoagulants) taken during hospitalization was determined. Fluorescence polarization immunoassay (FPIA) performed by an Abbott GmbH TDx automatic analyzer was used to determine the CBZ serum concentration. The limit of detection was 0.005 µg/mL. The therapeutic range for carbamazepine was established as 4–12 µg/mL according to the manufacturer’s requirements.

### 2.3. Statistics

The Polish version of the statistical software Statistica 12.0 (StatSoft, Tulsa, OK, USA) was used to calculate statistical parameters. The Shapiro–Wilk test assay revealed that the distribution of continuous variables did not meet the criteria of a normal distribution. As a result, quantitative variables were shown as medians and quartile ranges. To compare the medians of independent variables, the Mann–Whitney test, the Kruskal–Wallis test, and the multiple comparison test were performed. Qualitative variables were presented as the number of patients with a particular feature and as the percentage of the analyzed group. Comparison between qualitative variables was performed using χ^2^. Values of *p* < 0.05 were treated as statistically significant. Values of *p* ≥ 0.05 and <0.10 were treated as a trend towards statistical significance. Values of *p* ≥ 0.10, which were treated as not significant, were replaced with the abbreviation *ns* (not significant).

## 3. Results

In total, 710 cases were analyzed, including 339 men and 371 women. The clinical characteristics of the study population, including the parameters of their CBZ drug therapy, are presented in Table 1. The average age of the study population was 19, the average height was 168 cm, the average weight was 59 kg, and the average BMI (body mass index) was 22.49 kg/m^2^. The most common reason for hospitalization was epilepsy—94.6%. The average daily carbamazepine intake among the examined patients was 600 mg (in the form of oral tablets). In 3.1% of cases, it was a single dose; 66.1% took two daily doses; and 26.9% took at least three doses per day. The average carbamazepine concentration in the blood serum of the study population was 5.58 µg/mL and remained within the therapeutic range. The therapeutic range was present in 71.0% of cases. The average time from the beginning of the carbamazepine treatment to the measurement of the carbamazepine concentration in the blood serum was 180 days.

An analysis of the study population showed statistically significant differences between the carbamazepine concentration depending on the main reason for hospitalization (Table 2, Figure 1). A statistically significant correlation was demonstrated between epilepsy and suicide attempt (*p* < 0.001) and between suspected drug overdose and suicide attempt (*p* = 0.005). The average concentration of carbamazepine in patients with epilepsy was 5.52 µg/mL. In this group of patients, a concentration below the therapeutic range (<4 µg/mL) was observed in 167 patients (24.9%), a concentration at the therapeutic level (4–12 µg/mL) was found in 491 patients (73.1%), and a concentration in the range above the therapeutic level (≥12 µg/mL) was found in 14 patients (2.1%). Among patients with a suspected drug overdose, the median drug concentration was 7.24 µg/mL. In these patients, concentrations below the therapeutic range were found in 6 patients (26.1%), concentrations at the therapeutic level were observed in 10 (43.5%), and in 7 patients (30.4%) the therapeutic range was exceeded. In the case of patients who made a suicide attempt, the average concentration of carbamazepine was 21.0 μg/mL. In all these patients, only a CBZ concentration above the therapeutic range was recorded. In the study population (excluding patients with no dose data and who made a suicide attempt), regardless of the number of doses taken per day, the median concentration of CBZ was within the therapeutic range and did not differ statistically (Table 2). Among patients taking carbamazepine once daily, only 45.5% were within the therapeutic concentration range. In patients receiving the drug twice daily or more, the therapeutic concentration range was seen statistically significantly more frequently—72.5% and 75% (*p* = 0.035), respectively. There were no statistically significant differences in the measured drug level according to sex and age (Table 2).

In the study population (excluding patients with no dose data and who made a suicide attempt), statistically significant differences were found between the daily dose and the daily dose of carbamazepine per kilogram of body weight depending on the patients’ age (Table 3, Figure 2). The average daily dose of carbamazepine in patients between 3 and 17 years of age amounted to 450 mg (daily dose per kilogram of body weight—13.64 mg). However, in patients ≥18 years of age it was 800 mg and 10.43 mg/kg bw (*p* < 0.00001), respectively. There were also statistically significant differences between the daily dose of carbamazepine and the level of CBZ. A statistically significant difference was demonstrated between the level below the therapeutic range and the therapeutic level (*p* < 0.001), and between the level below and above the therapeutic level (*p* < 0.001). A trend towards statistical significance (*p* = 0.081) was also demonstrated between the therapeutic level and the level above the therapeutic window (Table 3). On average, the daily dose amounted to 400 mg in patients with a CBZ concentration below the therapeutic level, 600 mg in patients at the therapeutic level, and 850 mg in patients with levels above the therapeutic range. Statistically significant differences were also found between the daily dose of carbamazepine per kilogram of body weight and the CBZ level. Statistically significant differences were demonstrated between the level below the therapeutic range and the therapeutic level (*p* < 0.001). On average, in patients with CBZ concentrations below the therapeutic level, the daily dose per kilogram of body weight amounted to 9.92 mg; in patients with levels at the therapeutic level it was 12.86 mg; and in patients with levels above the therapeutic range it was 12.50 mg. The average daily dose of CBZ was 600 mg for both men and women. The average dose per kilogram of body weight was 11.43 mg in men and 12.1 mg in women. These results were not statistically significant.

In the study population (excluding patients with no dose data and who made a suicide attempt), statistically significant differences were found depending on the main reason for hospitalization and age (*p =* 0.003) (Table 4). In both age groups (3–17 and ≥18 years old), epilepsy was the dominant cause of hospitalization. In addition, there was a higher proportion of elderly patients with a suspected drug overdose and a suicide attempt. In terms of gender, epilepsy was also the main reason for hospitalization. There was a higher proportion of women with a suspected drug overdose and a much higher proportion of men with a suicide attempt, although the observed differences did not reach statistical significance. Statistically significant differences were found depending on age and the number of doses of the drug taken per day *(p* = 0.0100). Patients from both age groups most often took CBZ twice a day (among patients between 3 and 17 years of age the percentage was 73.3%, and after the age of 18 years it amounted to 64.7%). Younger people were more often treated with one dose a day (4.0% vs. 2.5%) but less frequently with at least 3 doses a day (22.7% vs. 32.8%). No statistically significant differences were found depending on sex and the number of doses taken per day. In the female population of the study group, there was a slightly higher frequency of patients reaching the therapeutic ranges (72.2% vs. 69.6%) and a lower frequency of patients exceeding the therapeutic (3.2% vs. 5.0%) and subtherapeutic levels (24.5% vs. 25.4%) than in the male population. However, these differences were not statistically significant. Additionally, the therapeutic range was more often observed in patients aged 3–17 years (73.5% vs. 68.8%) compared with the population ≥18 years of age, whereas the subtherapeutic levels were reached slightly less frequently (24.1% vs. 25.7%). On the other hand, the therapeutic level was exceeded over twice less frequently (2.4% vs. 5.6%). The obtained differences were close to statistical significance (*p* = 0.0082).

### Study Limitations

Several limitations of our study merit consideration—primarily, the incomplete data for the study population resulting mainly from the large number of subjects and the duration of the follow-up. Unfortunately, they were not included in the electronic database. Moreover, there was no information about potential pregnancy, additional pharmacotherapy, or comorbidities that affected the level of CBZ. In addition, it was not possible to verify the compliance declared by the patient.

## 4. Discussion

Carbamazepine is a first-generation anticonvulsant drug recommended for use in the treatment of focal epilepsy [1]. The main effect of the drug is to stabilize the cell membranes of nerve fibers, inhibit neuronal discharges, and reduce excitatory synaptic transmission [4]. In some clinical situations, however, the risk of significant drug interactions is a major limitation affecting the use of CBZ [4]. In the study population, the determination of the CBZ level was most often combined with consultation with a clinical pharmacologist. Therefore, TDM supports pharmacotherapy in certain cases. In our study, 71% of patients taking CBZ had drug concentrations at the therapeutic level. These results are slightly lower than those obtained by Shakya et al. [14], who reported that 79.3% of patients in a similar population also possessed therapeutic levels of CBZ. On the other hand, 24.9% of the studied population had concentrations below the therapeutic level, which could be related, for instance, to noncompliance with the medical recommendations or the use of other drugs that reduced the concentration of CBZ. Compliance can be improved by reducing the number of daily doses and the routine monitoring of the drug levels. In the study population, patients aged 3–17 years reached the therapeutic range more often (73.5% vs. 68.8%) compared to the population ≥18 years of age, whereas the subtherapeutic levels were observed slightly less frequently (24.1% vs. 25.7%). The obtained results are very important because, according to other studies, this age group has a higher risk of noncompliance with medical recommendations, which is the reason for the low plasma concentrations of CBZ [15,16,17]. In the study by Specht et al., a drop in serum levels >50%, indicating medication noncompliance, was noted in 44.3% of the seizures in young adults with epilepsy. It is worth noting that the therapeutic level was exceeded less often in the population between 3 and 17 years of age, despite their significantly higher dose per kilogram of body weight than the population ≥18 years (13.64 mg/kg vs. 10.43 mg/kg, *p* < 0.0001). These results confirm that in younger age groups there is a faster metabolism of CBZ associated with the approximately 20% higher activity of the CYP3A4 isoenzyme [18]. Among patients with a suspected drug overdose, patients ≥18 years of age were statistically significantly more frequent (73.9% vs. 26.1%), and suicide attempts were only made by elderly patients (100.0% vs. 0.0%*, p* = 0.003). At the same time, in the population ≥18 years of age, the dose per kilogram of body weight was significantly lower than in the group between 3 and 17 years (10.43 mg/kg vs. 13.64 mg/kg, *p* < 0.0001).

## 5. Conclusions

To sum up, the TDM results of CBZ were evaluated for both sexes in different age groups, where epilepsy was the predominant diagnosis (94.6%). The TDM results of CBZ showed that only 71% of all samples were at the therapeutic level (with little variation depending on age, sex, and the number of doses taken per day). In the study population, CBZ was most often taken twice a day (regardless of sex and age), which is in line with current dosing recommendations. The differences in CBZ levels may result from, among other things, different metabolic patterns dependent on sex and age. Pregnancy also affects CBZ concentration because an increased metabolism reduces its concentration. In order to ensure the maximum effectiveness and safety of the therapy, it is necessary to monitor the concentration of CBZ. TDM should be conducted in patients regardless of their sex and age, and the dosage should be adjusted to the results obtained, because the effectiveness of epilepsy treatment is strongly related to the CBZ levels in patients’ plasma [18,19].

For the above reasons, from the point of view of a clinical pharmacologist, TDM of CBZ is a standard procedure, similar to, e.g., TDM in digoxin therapy [12], in contrast to a therapy where TDM is available, but due to the high safety of the therapy it can be considered only in selected patients at increased risk of complications or drug interactions [20].

## Figures and Tables

**Figure 1 jcm-10-05396-f001:**
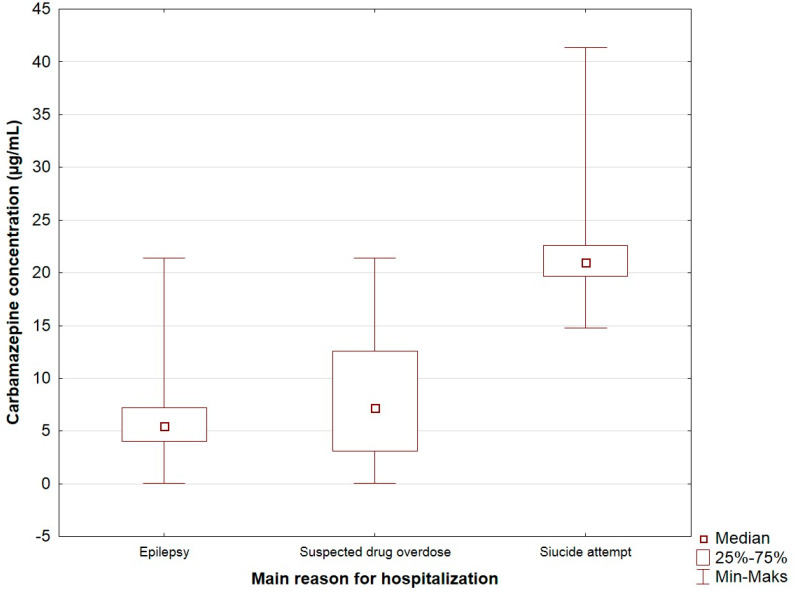
The carbamazepine concentration (µg/mL) depending on the main reason for hospitalization (marker—median; edges of the box—upper and lower quartile; whiskers—range of non-outlier values).

**Figure 2 jcm-10-05396-f002:**
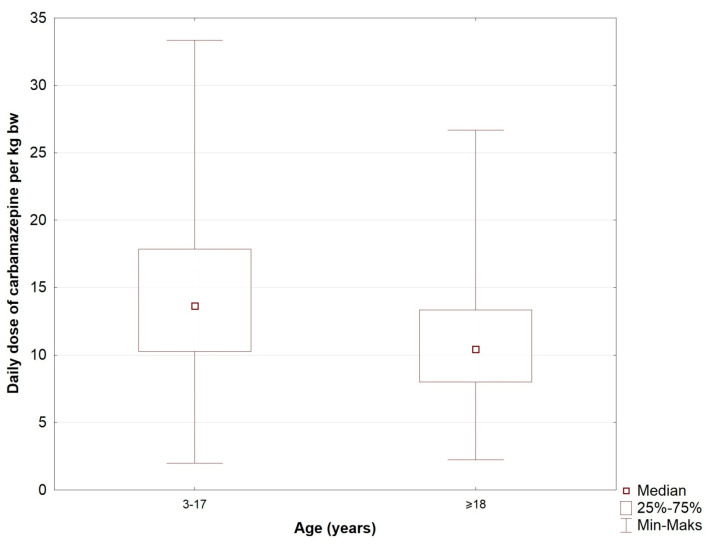
Daily dose of CBZ (mg) depending on age in the study population (marker—median; edges of the box—upper and lower quartile; whiskers—range of non-outlier values).

**Table 1 jcm-10-05396-t001:** Clinical characteristics of the study population (median (lower quartile–upper quartile) or number (percent)).

Study Feature (*n* = 710)	Property Value
Age	19.0 (11.0–31.0)
3–17 (years)	332 (46.8%)
≥18 (years)	378 (53.2%)
Height (cm)	168.0 (160.0–175.0)
Body weight (kg)	59.0 (35.0–70.0)
BMI (kg/m^2^)	22.49 (19.59–25.35)
Sex	
Men	339 (47.7%)
Women	371 (52.3%)
The main reason for hospitalization	
Epilepsy	672 (94.6%)
Suspected drug overdose	23 (3.2%)
Suicide attempt	8 (1.1%)
Not specified	7 (1.0%)
The daily dose of carbamazepine (mg) *	600.0 (400.0–800.0)
The daily dose of carbamazepine per kilogram of body weight (mg) *	12.00 (9.09–15.79)
The number of doses per day	
1	22 (3.1%)
2	469 (66.1%)
≥3	191 (26.9%)
No data (including patients with suicide attempt)	28 (3.9%)
Carbamazepine concentration (µg/mL) Time period from the beginning of treatment with carbamazepine (days)	5.58 (4.01–7.33)
	180.0 (60.0–700.0)
Carbamazepine below the therapeutic level (<4 µg/mL)	177 (24.9%)
The therapeutic level of carbamazepine (4–12 µg/mL)	504 (71.0%)
Carbamazepine above therapeutic level (≥12µg/mL)	29 (4.1%)

Abbreviations: BMI—body mass index. * patients with a suicide attempt and no dose data were excluded.

**Table 2 jcm-10-05396-t002:** Carbamazepine concentration in the study population depending on selected clinical parameters (median (lower quartile–upper quartile) or number [percent]).

Study Feature	Carbamazepine Concentration (µg/mL)	*p*
The main reason for hospitalization (*n* = 703)		<0.0001
Epilepsy	5.52 (4.01–7.22)	
Suspected drug overdose	7.24 (3.08–12.61)	
Suicide attempt	21.0 (19.67–22.63)	
Sex (*n* = 710)		0.408
Women	5.52 (4.01–7.24)	
Men	5.63 (3.88–7.40)	
Age (years) (*n* = 710)		0.347
3–17	5.50 (4.07–7.14)	
≥18	5.76 (3.92–7.54)	
The number of doses per day (*n* = 682)		0.2830
1	5.03 (3.0–6.48)	
2	5.70 (3.98–7.26)	
≥3	5.36 (4.14–7.23)	

**Table 3 jcm-10-05396-t003:** Pharmacotherapy with carbamazepine in the study population depending on selected clinical parameters (median (lower quartile–upper quartile) or number (percent)).

Study Feature (*n* = 682)	The Daily Dose of Carbamazepine (mg)	*p*	The Daily Dose of Carbamazepine Per Kilogram of Body Weight (mg)	*p*
Sex		0.3593		0.0891
Men	600 (400–800)		11.43 (8.57–15.38)	
Women	600 (400–800)		12.1 (9.52–16.0)	
Age		<0.00001		<0.00001
3–17(years)	450 (300–600)		13.64 (10.26–17.86)	
≥18 (years)	800 (600–900)		10.43 (8.00–13.33)	
Carbamazepine below the therapeutic level (<4 µg/mL)	400 (300–600)	<0.00001	9.92 (6.78–12.50)	<0.00001
The therapeutic level of carbamazepine (4–12 µg/mL)	600 (450–800)		12.86 (9.68–16.67)	
Carbamazepine above the therapeutic level (≥12µg/mL)	850 (600–1000)		12.50 (10.0–18.46)	

**Table 4 jcm-10-05396-t004:** Daily carbamazepine dose and the main reason for hospitalization of the study population depending on age and sex (median (lower quartile-upper quartile) or number (percent)).

Study Feature	Sex		*p*	Age (years)		*p*
	Men	Women		3–17	≥18	
**The main reason for hospitalization (*n* = 703)**						
Epilepsy	320 (47.6%)	352 (52.4%)	0.056	325 (48.4%)	347 (51.6%)	0.003
Suspected drug overdose	9 (39.1%)	14 (60.9%)		6 (26.1%)	17 (73.9%)	
Suicide attempt	7 (87.5%)	1 (12.5%)		0 (0.00%)	8 (100%)	
**The number of doses per day (*n* = 682)**						
1	14 (63.6%)	8 (36.4%)	0.139	13 (4.0%)	9 (2.5%)	0.0100
2	213 (45.4%)	256 (54.6%)		236 (73.3%)	233 (64.7%)	
≥3	97 (50.8%)	94 (49.2%)		73 (22.7%)	118 (32.8%)	
**Carbamazepine concentration (µg/mL)**						
Carbamazepine below the therapeutic level (<4 µg/mL)	86(25.4%)	91(24.5%)	0.4502	80(24.1%)	97(25.7%)	0.0082
The therapeutic level of carbamazepine (4–12 µg/mL)	236(69.6%)	268(72.2%)		244(73.5%)	260(68.8%)	
Carbamazepine above the therapeutic level (≥12 µg/mL)	17(5.0%)	12(3.2%)		8.0(2.4%)	21.0(5.6%)	

## Data Availability

The study data may be available on request.

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
