# Peer review of "Therapeutic Drug Monitoring of Carbamazepine: A 20-Year Observational Study"

_jcm, 2021, doi:10.3390/jcm10225396_

Round 1

Reviewer 1 Report

The paper analyzes a large population of hospitalized patients, who underwent a therapeutic drug monitoring with carbamazepine for 20 years. This is a great job of compiling data obtained from a database and, partly due to lack of follow-up and / or due to the lack of medical history, information that could be valuable is unknown, for example, pregnancy, medication and pathology concomitant. The authors mention this limitation of the study.

Having clarified this important aspect, however, the analysis carried out by the authors obtained important and interesting results for the general medical community, since the drug studied, in addition to being a widely used antiepileptic, is used by several specialties for pain treatment.

Some observations: The introduction should not include the separation of the paragraph 'Epilepsy', where very general aspects of the disease are commented, perhaps it would be more useful to remember the precise indications of carbamazepine in the different focal epileptic syndromes, as a first-line drug, according to the ILAE classification of 2017 and add some level of evidence for its use.

On line 117 the sentence starts with a number, I suggest writing it.

Some considerations in the discussion are too speculative, partly due to the lack of commented data.

Author Response

Journal of Clinical Medicine

Author's Response to Decision Letter for (jcm-1455233).

Title: Therapeutic drug monitoring of carbamazepine: a 20-year observational study.

We would like to thank to the reviewer for all advices and effort. All suggestions have been implemented point by point.

  1. Introduction: has been improved accordingly to the suggestions (indication of carbamazepine in the different focal epileptic syndromes, as a first-line drug, according to the ILAE classification of 2017). Separation of the paragraph 'Epilepsy' has been deleted.
  2. Results: on line 117 the sentence has been improved accordingly to the suggestions ("Seven hundred and ten cases were analyzed, including 339 men and 371 women").
  3. Discussion: has been improved accordingly to the suggestions ("In a study by Specht et al. a drop in serum levels >50% indicating medication noncompliance was noted in 44.3% of the seizures in young adults with epilepsy").

Conclusion: have been added accordingly to suggestion.

Reviewer 2 Report

The authors reported on the blood level of carbamazepine (CBZ). The research was based on a retrospective analysis including 710 ambulatory and hospitalized patients treated with CBZ between the years 1991-2011. The method used for determination of CBZ concentration was fluorescence polarization immunoassay. The therapeutic range of CBZ was set 4-12 ug/ml. They found that that 71% of all samples were at the therapeutic level. The authors reported that patients aged 3-17 years reached the 252 therapeutic range more often (73.5% vs. 68.8%) compared to the population ≥ 18 years of 253 age. They also provided the CBZ level in patients with overdose and suicide attempt.  

 Their data is large enough to gain some insight in the real-world epilepsy treatment situation. The finding may be helpful to the pharmacist and other health care providers who takes care of patients with epilepsy.

Author Response

Journal of Clinical Medicine

Author's Response to Decision Letter for (jcm-1455233).

Title: Therapeutic drug monitoring of carbamazepine: a 20-year observational study.

  We would like to thank to the reviewer for all advices and effort.